# Beyond the Indicators: Improving Science, Scholarship, Policy and Practice to Meet the Complex Challenges of Sustainability

**Ortwin Renn** [1] , **Ilan Chabay** [1] , **Sander van der Leeuw** [2] **and Solène Droy** [1],*

[1]  Institute for Advanced Sustainability Studies, 14467 Potsdam, Germany;
ortwin.renn@iass-potsdam.de (O.R.); ilan.chabay@iass-potsdam.de (I.C.)

[2]  Arizona State University, Global Biosocial Complexity Initiative, Tempe, AZ 85281-2701, USA;
vanderle@asu.edu

*  Correspondence: solene.droy@iass-potsdam.de; Tel.: +49-1-713135987

**Abstract:** Many teams have developed a wide range of numerical or categorical indicators of progress in the implementation of the SDG targets. But these indicators cannot identify why target goals have not been accomplished, whether or how they do or do not do justice to the social and cultural context in which they are applied, and how newly emerging social dynamics affect indicators. Nor do they provide means for resolving conflicting values and making balanced trade-offs. Our starting point in examining why we have not been successful in progressing towards sustainability is that the sustainability conundrum is primarily a societal, rather than an environmental problem. Our present emphasis is to maintain our way of life while minimizing its impact, hoping that such a minimization strategy would make the world more sustainable. Reducing for example the extent of pollution but keeping the same industries alive would not be sufficient for a transformation towards sustainability. Instead we should ask "How did we come to this point and what practices, in our societies and in our science, need to change to make progress towards sustainability?" To answer these questions, one needs to go much further back than usual in the history of western societies to identify the societal, scientific, technological and environmental co-evolutionary dynamics that have brought us to the current conundrum. And the fact that most societal challenges are of the "wicked" kind, as well as the need to decide among many societal options and many future pathways that may lead to positive results require that we seriously engage in using "Complex Systems" approaches. It is up to our scientific community to identify these pathways, and we need to move fast!

**Keywords:** sustainability indicators; UN SDGs; sustainability science; conflicting societal values; evidence-informed judgments; complex systems science; narrative expressions

---

## 1. Insufficiency of Sustainability Indicators

The 2015 agreement setting forth the UN Agenda 2030 for Sustainable Development Goals (SDGs) is an important achievement that poses complex and demanding challenges. These are inherent in the interdependent nature of the goals and the great rapidity of social, economic, and bio-physical changes called for [1,2]. To adequately address them, judgments must determine contextually and culturally appropriate balances between independently valuable, but often conflicting targets [3]. Simultaneously, policies must ensure global coherence across local and regional actions, so that local efforts do not destructively interfere with each other, nor overstep limitations in the resources of the planet [4]).

This Comment, as a brief concept paper, summarizes a set of ideas for strengthening the capacity and processes of societies in general and academia in particular to address the complex challenges

of sustainability. The points raised in this paper are intended to stimulate substantive dialogues and actions toward just and equitable sustainable futures. A longer, more detailed discussion of these points, which were developed by a group of 17 sustainability science experts in the first Global Strategy Sustainability Forum (GSSF) (https://www.iass-potsdam.de/en/research/global-sustainability-strategy-forum), is currently in review by one of the prime journals in sustainability studies. It reflects an assessment of the current state of sustainability research and argues for appropriate steps toward improvement in academia as well as in the modes of cooperation between academia, the corporate sector, political institutions and civil society.

The core of our argument is that many teams have developed a wide range of numerical or categorical indicators of progress in the implementation of the SDG targets (e.g., [5–8]). But these indicators cannot identify *why* target goals have not been accomplished, *whether* or *how* they do or do not do justice to the social and cultural context in which they are applied, and *how* newly emerging social dynamics affect indicators. Nor do they provide means for resolving conflicting values and making balanced trade-offs [9–11].

## 2. Reconsidering Sustainability as a Societal Challenge

Our starting point in examining why we have not been successful in progressing towards sustainability is that the sustainability conundrum is primarily a societal, rather than an environmental problem. The overbearing emphasis on reducing greenhouse gas emissions, and the wide range of research on issues, such as the future water and food security, shows a clear bias towards maintaining our way of life while minimizing its impact, hoping that such a minimization strategy would make the world more sustainable [12–14]. Increasingly, awareness is growing that reducing, for example, the extent of air pollution or of water use, but keeping the same industries alive by promoting different technologies, would not be sufficient for a transformation towards sustainability. Nor does such research deal with the core dynamics that have driven our societies towards the current conundrum. Instead we should ask "how did we come to this point and what practices, in our societies and in our science, need to change to make substantive and rapid progress towards sustainability?"

To answer these questions, one needs to go much further back than usual in the history of our western societies to identify the societal, scientific, technological and environmental co-evolutionary dynamics that have brought us to the current conundrum. Transformations in European thinking in the 14th century place the roots for our distinction between "nature" and "culture" [15]; globalization has its roots in the 16th and 17th century global explorers and the transformation of European trading empires in the 1800s into resource exploitation colonies; the Industrial Revolution, by reducing the cost of energy, transformed global connectivity and led to the explosion of technological innovations, etc.

And the fact that most societal challenges are of the "wicked" kind [16], as well as the need to decide among many societal tradeoffs and many future pathways that may or may not lead to positive results [17], require that we seriously engage in using "Complex Systems" approaches. Whilst there are efforts in this direction, many of those are still only partial [18–21]. It is up to our scientific community to identify these pathways, and we need to move very quickly!

To attain the desired goals, we need to focus on sprints to feasible near-future goals and concentrate on regional contexts, while keeping the global requirements in mind. As we have seen in the Conferences of the Parties in the years up to 2015, being fixated on the global scale slows down the process, risks glossing over local and regional differences in context and culture, and removes incentives and agency from many actors [22]. We will face only resistance if we try to prescribe uniform "solutions" for all instances: for example, that all nations must phase out fossil fuels or become vegetarians. Global needs are best met by implementation that takes local and regional circumstances and cultures into account. We need therefore to be aware of the special circumstances in each region and assess the potential for change in partnerships with the local population. If we are, solutions can be local without losing the interlinkages between different regions, or with the global challenges. Lenschow et al. [23] argue this cogently by emphasizing "telecoupling" in environmental politics. The main argument is that

each regional situation requires its unique approach but from each case other regions can learn and come up with their own solutions informed by best practice and successful experience. Persistently progressing step by step in this manner in all regions of the world will have more impact than waiting for the big turnaround. It will also provide opportunities to reflect on achievements, new challenges and opportunities, and learn how to approach them in a constructive, respectful and peaceful manner, while reducing unintentional effects.

## 3. Reassessing the Role of Science and Scholars to Adequately Address the Challenges

This brings us to the crucial question of what strategic advice for assessing policy options and their likely impacts can social and natural science offer, in particular with respect to sustainable futures and designing necessary policy reforms [24]. By providing orientation and anticipating results and unintended consequences [25] of policies, science can help in making informed, prudent judgments about threats that are complex, difficult to comprehend, and often stochastic in nature [26]. Integrating science-informed ideas in such decisions is complex and requires finding a delicate balance between hierarchical, market-based and network-based approaches that provide the conceptual foundation for governance structures and procedures [27]. Scientists can assist local communities and nations to assess the likelihood of consequences of actions that they are considering. Through active collaboration and communication with members of the affected communities, scientists can co-develop options and priorities for actions. Where possible, the people affected will in the end decide for themselves [28,29].

In addition to providing scientific evidence in the form of statistics, models and simulations relevant to informing (not determining) action, scientists should be engaged in gathering, understanding, and, where useful, generating narratives together with the people affected and integrating these into education and policy making as powerful elements for sense-making [30]. In that process, adopting a complex systems-based approach that focuses on learning *from* the past *about* the present and *for* the future is more helpful than explaining the present in terms of chains of cause-and-effect [31]. That will widen our insight in the high-dimensional socio-environmental systems involved and enable us to include multiple possible futures in our counsel. In addition, in the complex socio-ecological systems in which humanity is embedded, causality often is a casualty of the consequences of complexity because, while we may observe correlations, linear connections between cause and effect often may not be possible.

Especially in the face of unknown, uncertain futures, undertaking substantial changes in behavior involve unfamiliar actions, learning new roles in society, and difficult transitions to changing norms and narratives. Scientific inquiries will not answer such questions. The answers are value-laden, implicitly or explicitly interwoven in narratives that sustain people's social identities. Such narratives are more effective if they are co-constructed with diverse sectors of society and if they present positive visions anticipating a future in which people have fulfilling roles and responsibilities, rather than being told as horror stories or doomsday scenarios [32].

Integrating scientific knowledge in culturally meaningful visions of the future provides the bridge between abstract scientific models about the effects of human interactions with their environments and the cultural narratives that create identity and a sense of belonging [33]. Making the benefits of transformation understandable, comprehensible and reasonable to all involved and presenting ideas to support those negatively affected by the changes will ease the transformation process.

## 4. Designing for Long-Term Desirable Change

As much as science needs to include plausible narratives to link to policy and collective action, policy itself needs to be evidence-informed [34]. Until now, much scientific evidence produced has been the result of curiosity-driven research characterized by different theories and methodologies. For effective policy implementation, that must be complemented by vision-driven, co-evolutionary and integrative evidence that achieves set goals through strategically construed science based on plural knowledge resources to guide action [35,36]. For example, there are many building traditions in

countries likely to suffer from increased climate change that include built-in cooling systems or fresh air venting devices based upon indigenous knowledge and proven practice, which could be optimized by scientific and technical innovations.

In the current rapid and encompassing changes in all domains of society, a more pro-active view needs to be developed, which includes potential futures that cannot be linearly derived from the present dynamics. Science (in the widest sense) needs to move from designing for stability to designing for long-term desirable change. We must learn how to do so with new conceptual and operational methods, including commensurate changes in educational institutions and approaches across ages and societies.

In conclusion, to engage effectively and meaningfully with other sectors of society to catalyze novel pathways to far-reaching societal transformations requires a profound change in the scientific approaches addressing transitions to sustainability. This means focusing on the societal dynamics driving our societies into an unsustainable future, rather than solely on mitigating the environmental consequences. These conclusions reflect the recommendations of a larger group of concerned sustainability experts (Global Sustainability Strategy Forum) who convened under the moderation of the authors for a week of intensive deliberation in March of 2019 (https://www.iass-potsdam.de/en/research/global-sustainability-strategy-forum). That international Forum identified substantial changes that are required in science and education and emphasized the need to engage in joint narrative constructions with diverse sectors of society in future forums and workshops on practice and policy. By analyzing the drivers behind the indicators, implementing a step by step, regional and bottom-up strategy, we are not abandoning the global perspective, but "thinking locally and acting locally" with full awareness of the global impacts in a world in which all the spheres of life are interconnected.

**Author Contributions:** Conceptualization, O.R., I.C., S.v.d.L., S.D.; methodology, O.R., I.C., S.v.d.L., S.D.; validation, O.R., I.C., S.v.d.L., S.D.; formal analysis, O.R., I.C., S.v.d.L., S.D.; investigation, O.R., I.C., S.v.d.L., S.D.; resources, O.R., I.C., S.v.d.L., S.D.; writing—original draft preparation, O.R., I.C., S.v.d.L., S.D.; writing—review and editing, O.R., I.C., S.v.d.L., S.D.; supervision, O.R., I.C., S.v.d.L.; project administration, S.D.; funding acquisition, O.R., S.D. All authors have read and agreed to the published version of the manuscript.

**Funding:** This research was funded by the VolkswagenStiftung Germany. The APC was funded by the VolkswagenStiftung Germany.

**Conflicts of Interest:** The authors declare no conflict of interest.

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
