# Peer review of "Beyond the Indicators: Improving Science, Scholarship, Policy and Practice to Meet the Complex Challenges of Sustainability"

_sustainability, doi:10.3390/su12020578_

Round 1
Reviewer 1 Report
It's very hard to review an article. It is very short. It is divided into 4 parts. However, their length does not allow any conclusions to be drawn. The authors do not clearly indicate the purpose of the study. No research methods have been used. The authors only give their own thoughts, rarely referring to literature. In my opinion, the article needs general improvement. The project's goal and research methods should be clearly stated. And also the results of own research should be shown. The article cannot be published in its current form. At the moment it is not a scientific article. The authors treated the analyzed topic very superficially.
Author Response
Comment 1:
It's very hard to review an article. It is very short. It is divided into 4 parts. However, their length does not allow any conclusions to be drawn. The authors do not clearly indicate the purpose of the study. No research methods have been used. The authors only give their own thoughts, rarely referring to literature. In my opinion, the article needs general improvement. The project's goal and research methods should be clearly stated. And also the results of own research should be shown. The article cannot be published in its current form. At the moment it is not a scientific article. The authors treated the analyzed topic very superficially.
Response 1:
Thank you very much for this comment.
There might have been a misunderstading on the purpose and intent of the present paper, here is an Explanation that should clarify our intent:
This Comment, as a brief concept paper, summarizes a set of ideas for strengthening the capacity and processes of societies in general and academia in particular to address the complex challenges of sustainability. The points raised in this paper are intended to stimulate substantive dialogues and actions toward just and equitable sustainable futures. A longer, more detailed discussion of these points, which were developed by a group of 17 sustainability science experts in the first Global Strategy Sustainability Forum (GSSF)[1], is currently in review by one of the prime journals in sustainability studies. It reflects an assessment of the current state of sustainability research and argues for appropriate steps toward improvement in academia as well as in the modes of cooperation between academia, corporate sector, political institutions and civil society.
We have added substantial references to literature. The bibliography has been improved. (Please see new manuscript).
[1] https://www.iass-potsdam.de/en/research/global-sustainability-strategy-forum
Reviewer 2 Report
This commentary piece aims to draw out potential linkages between natural and social sciences, and between science and policy. It has compiled many key ideas on how sustainability research should advance. The arguments are structured very well and logically presented in four sections, one building on another. Its intention of communicating the consensus of ‘concerned sustainability experts’ is justified and admirable.
However, I must admit that I struggle to find sufficient novelty that would compel me as a researcher to read through this short commentary piece. In addition, I would prefer to see stronger support of your arguments even though this may be a commentary piece rather than a research article.
Here are my specific comments:
Line 29-32
While this claim makes a lot of sense, has there been any discourse analysis on sustainability governance that supports your claim?
Line 34-36
This is a highly generic claim on which no one will disagree. However, the readers benefit from a couple of lines of more specific observations from big history research for example. The role of institutions in accelerating the convergence of the trends may be a useful reference.
Line 39
I absolutely agree with the authors that the complex systems approach is the way to go. However, research and policy practitioners are much more ready to apply this approach today. There are tools designed to model the interlinkages between the indicators but the authors have not made references to them.
Line 44-49
The authors seem to depict the SDGs agenda as a top-down agenda that prescribes a set of universal solutions. There has been a strong promotion of certain kinds of solutions, but I do not see these solutions enshrined in the Agreement. I also do not see clear sanctions administered for not implementing these solutions. The solutions are promoted as one of the options, and SDGs agenda, in fact, has been criticized for allowing nations to ‘cherry-pick’ the indicators and the associated means of implementation. The idea of common but differentiated responsibilities should counter the authors’ claim on the rigid and context-free prescription of global solutions.
Line 59-61
‘The people affected are best placed and should have the opportunity to decide for themselves.’ This claim may be emotive to make, but may not apply in many cases. People are easily swayed and would fall into the trap of cognitive biases. Brexit is an illustrious example whereby once exposed to the consequences, voters can change their minds quickly. There is also the potential danger of arguing against any reliance on experts. I wonder if the way out of this false dichotomy is how a hybrid governance situation can make good use of hierarchy, network and market modes.
Line 71-78
Indeed, stories communicate the nature of identities, and how they influence preferences much more effectively than the indicators alone. I do not think this adds much to how we already think. It may be more valuable to point out how indicators and narratives can be integrated in a way that brings out the best in the two of them. Unfortunately, the authors have not offered such a vision until the last line of the Commentary (Line 108-111). However, the rest of the commentary should be aligned with this understanding.
All in all, I thought if the Journal is looking for a compilation of easy-to-strike consensus among the sustainability experts. However, if the Journal would like to engage the readers in concrete and novel improvements to the emerging ‘pluripotent’ sustainability discourse, then I would suggest the authors to perhaps include more complicated examples of governance practice. It is also probable that it is not possible to achieve transformational consensus around reforming the governance regime among the sustainability experts, in the same way that such global consensus in the policy world is too prescriptive, slow and context-ignorant. Incrementalism is all that we can afford in both arenas.
Author Response
Comment 1:
However, I must admit that I struggle to find sufficient novelty that would compel me as a researcher to read through this short commentary piece. In addition, I would prefer to see stronger support of your arguments even though this may be a commentary piece rather than a research article.
Response 1:
Thank you very much for your comments. Indeed, this is a commentary piece rather than a Research article. We have improved it thanks to your suggestions. We have added substantial references that can address your concerns (see new manuscript). It is worth mentioning that this Comment, as a brief concept paper, summarizes a set of ideas for strengthening the capacity and processes of societies in general and academia in particular to address the complex challenges of sustainability. The points raised in this paper are intended to stimulate substantive dialogues and actions toward just and equitable sustainable futures. A longer, more detailed discussion of these points, which were developed by a group of 17 sustainability science experts in the first Global Strategy Sustainability Forum (GSSF)[1], is currently in review by one of the prime journals in sustainability studies. It reflects an assessment of the current state of sustainability research and argues for appropriate steps toward improvement in academia as well as in the modes of cooperation between academia, corporate sector, political institutions and civil society.
[1] https://www.iass-potsdam.de/en/research/global-sustainability-strategy-forum
Comment 2:
While this claim makes a lot of sense, has there been any discourse analysis on sustainability governance that supports your Claim?
Response 2:
We have added literature references on this.
Comment 3:
This is a highly generic claim on which no one will disagree. However, the readers benefit from a couple of lines of more specific observations from big history research for example. The role of institutions in accelerating the convergence of the trends may be a useful reference.
Response 3:
We have integrated some examples from big history research.
Response 4:
I absolutely agree with the authors that the complex systems approach is the way to go. However, research and policy practitioners are much more ready to apply this approach today. There are tools designed to model the interlinkages between the indicators but the authors have not made references to them.
Comment 4:
Thank you. Whilst there are efforts in this direction, many of those are still only partial (Preiser et al. 2018; Schill et al. 2019; Reyers et al. 2018; Schlüter et al. 2019).
Comment 5:
The authors seem to depict the SDGs agenda as a top-down agenda that prescribes a set of universal solutions. There has been a strong promotion of certain kinds of solutions, but I do not see these solutions enshrined in the Agreement. I also do not see clear sanctions administered for not implementing these solutions. The solutions are promoted as one of the options, and SDGs agenda, in fact, has been criticized for allowing nations to ‘cherry-pick’ the indicators and the associated means of implementation. The idea of common but differentiated responsibilities should counter the authors’ claim on the rigid and context-free prescription of global solutions.
Response 5:
The reviewer links this to the SDG’s directly, but it addresses the much wider field of “global warming” research. The one “global” aspect of the SDG’s is that that effort is rooted in a “progress” idea. But many people see “their” solution as applicable to, and necessary for, the whole world.
We need to be aware of the special circumstances in each region and assess the potential for change in partnerships with the local population. Persistently progressing step by step independently in all regions of the world will have more impact than waiting for the big turnaround. It will also provide opportunities to reflect on achievements, new challenges and opportunities, and learn how to approach them in a constructive, respectful and peaceful manner, while reducing unintentional effects.
Comment 6:
The people affected are best placed and should have the opportunity to decide for themselves.’ This claim may be emotive to make, but may not apply in many cases. People are easily swayed and would fall into the trap of cognitive biases. Brexit is an illustrious example whereby once exposed to the consequences, voters can change their minds quickly. There is also the potential danger of arguing against any reliance on experts. I wonder if the way out of this false dichotomy is how a hybrid governance situation can make good use of hierarchy, network and market modes.
Response 6:
This brings us to the crucial question of what strategic advice for assessing policy options and their likely impacts can social and natural science offer, in particular with respect to sustainable futures and designing necessary policy reforms (Renn 2019). By providing orientation and anticipating results and unintended consequences (Nowotny 2017) of policies, science can help in making informed, prudent judgments about threats that are complex, difficult to comprehend, and often stochastic in nature (Lucas et al. 2018). Integrating science-informed ideas in such decisions is complex and requires finding a delicate balance between hierarchical, market-based and network-based approaches that provide the conceptual foundation for governance structures and procedures. Scientists can assist local communities and nations to assess the likelihood of consequences of actions that they are considering. Through active collaboration and communication with members of the affected communities, scientists can co-develop options and priorities for actions. Where possible, the people affected will in the end decide for themselves (Dryzek, 1996; Oburu and Yoshikawa, 2018).
Comment 7:
Indeed, stories communicate the nature of identities, and how they influence preferences much more effectively than the indicators alone. I do not think this adds much to how we already think. It may be more valuable to point out how indicators and narratives can be integrated in a way that brings out the best in the two of them. Unfortunately, the authors have not offered such a vision until the last line of the Commentary (Line 108-111). However, the rest of the commentary should be aligned with this understanding.
Response 7:
Thank you. It is difficult to understand clearly what the reviewer says about this paragraph, and how (s)he wants the whole paper to be aligned to the integration of indicators and narratives.It is more about integrating scientific knowledge in culturally meaningful visions of the future. We have substantially improved the last paragraph on the role of narratives.
Round 2
Reviewer 1 Report
In my opinion, the level of the article is low. It contains only theoretical considerations. It lacks any own research. Only desk research was carried out.
Author Response
Many thanks for this comment.
There seems to be a mismatch between our purpose in writing and the reviewer’s expectation.
This Comment, as a brief concept paper, summarizes a set of ideas for strengthening the capacity and processes of societies in general and academia in particular to address the complex challenges of sustainability. The points raised in this paper are intended to stimulate substantive dialogues and actions toward just and equitable sustainable futures. A longer, more detailed discussion of these points, which were developed by a group of 17 sustainability science experts in the first Global Strategy Sustainability Forum (GSSF)[1], is currently in review by one of the prime journals in sustainability studies. It reflects an assessment of the current state of sustainability research and argues for appropriate steps toward improvement in academia as well as in the modes of cooperation between academia, corporate sector, political institutions and civil society.
To engage effectively and meaningfully with other sectors of society to catalyze novel pathways to far-reaching societal transformations requires a profound change in the scientific approaches addressing transitions to sustainability. This means focusing on the societal dynamics driving our societies into an unsustainable future, rather than on mitigating the environmental consequences. These conclusions reflect the recommendations of a larger group of concerned sustainability experts (Global Sustainability Strategy Forum) who convened under the moderation of the authors for a week of intensive deliberation in March of 2019.
[1] https://www.iass-potsdam.de/en/research/global-sustainability-strategy-forum
Thank you
Reviewer 2 Report
The Comment piece has been significantly improved, especially on what motivates it in the first place. Most importantly, it is a reaction to the dominant discourse at a particular conference – the Global Strategy Sustainability Forum (GSSF).
Given that the authors’ core argument relates to the over-reliance on indicators to describe progress, you may like to delve into the Global Policy - Special Issue on Knowledge and Politics in Setting and Measuring SDGs. Fukuda-Parr’s introductory article summarises more concrete and specific suggestions on improving the over-reliance issue. Perhaps the authors would like to reflect on the limitations of drawing insights from the 17 experts who attended the GSSF. What could they have missed given their areas of expertise? Are their views really representative of the broader community of sustainability experts? These questions are prompted by the observation that seminal pieces from sustainability experts like Frank Biermann. On the history of western societies (see Line52 – 59), it may be helpful to integrate the understanding of anthropocene through big history lens. David Christian’s most recent chapter on The Anthropocene Epoch in The Oxford Illustrated History of the World may be an appropriate reference. On contextualization (see Line 66-77), the false dichotomy of global and local solutions may not be helpful. Perhaps the middle way involves the concept of telecoupling which sees the interlinkages between spatially distant regions which may face similar global challenges but can co-evolve local solutions. The authors stated that “Persistently progressing step by step independently in all regions of the world”. The point of telecoupling is that solutions can be local not at the expense of losing the interlinkages. See Lenschow’s 2015 article on telecoupling in Environmental Politics. For this statement - “requires finding a delicate balance between hierarchical, market-based and network-based approaches that provide the conceptual foundation for governance structures and procedures”, the authors could reference works on hybrid governance in SDGs such as Louis Meuleman’s Metagovernance for Sustainability (2018) and Ryan Wong’s Balancing Institutions (2019). The authors might like to add a reference or two to Line 98-101, Line 121-125.This piece offers clarity around the current consensus on sustainability governance research, rather than novel insights or theoretical innovations. In academic discourse, it is highly valuable to take stock and gather momentum again. This much improved Comment piece will be ready for publication once the authors have ensured that their main arguments are appropriately referenced. Thank you for this meaningful exchange.
Author Response
Thank you very much for this very helpful review and fruitful Exchange.
We have taken the reviewer's suggestions into account, especially on contextualisation and the concept of telecoupling and the reference to Lenschow's work, as well as on hybrid governance and the reference to Louis Meuleman's work.
We have substantially edited the Paragraph on contextualisation (lines 89 to 106).
Thank you again for a very constructive Revision process.